# Naringenin Induces ROS-Mediated ER Stress, Autophagy, and Apoptosis in Human Osteosarcoma Cell Lines

**DOI:** 10.3390/molecules27020373

**Published:** 2022-01-07

**Authors:** Chiang-Wen Lee, Cathy Chia-Yu Huang, Miao-Ching Chi, Kuan-Han Lee, Kuo-Ti Peng, Mei-Ling Fang, Yao-Chang Chiang, Ju-Fang Liu

**Affiliations:** 1Department of Nursing, Division of Basic Medical Sciences, Chronic Diseases and Health Promotion Research Center and Research Center for Chinese Herbal Medicine, Chang Gung University of Science and Technology, Puzi City 61363, Taiwan; cwlee@gw.cgust.edu.tw; 2Department of Orthopaedic Surgery, Chang Gung Memorial Hospital, Puzi City 61363, Taiwan; mr3497@cgmh.org.tw; 3Department of Safety Health and Environmental Engineering, Ming Chi University of Technology, New Taipei City 243303, Taiwan; 4Department of Life Sciences, National Central University, Taoyuan City 320317, Taiwan; cathy80787@gmail.com; 5Department of Respiratory Care, Chang Gung University of Science and Technology, Puzi City 613, Taiwan; mcchi@gw.cgust.edu.tw; 6Division of Pulmonary and Critical Care Medicine, Chang Gung Memorial Hospital, Kaohsiung 833, Taiwan; anti0822@hotmail.com; 7Department of Pharmacy, Chia Nan University of Pharmacy and Science, Tainan 71710, Taiwan; 8Center for Environmental Toxin and Emerging-Contaminant Research, Cheng Shiu University, Kaohsiung 833, Taiwan; k6764@gcloud.csu.edu.tw; 9Super Micro Research and Technology Center, Cheng Shiu University, Kaohsiung 833, Taiwan; 10Department of Medical Research, China Medical University Hospital, China Medical University, Taichung 40402, Taiwan; 11School of Oral Hygiene, College of Oral Medicine, Taipei Medical University, Taipei 11031, Taiwan

**Keywords:** osteosarcoma, naringenin, ROS, ER stress, autophagy, apoptosis

## Abstract

Osteosarcoma, a primary bone tumor, responds poorly to chemotherapy and radiation therapy in children and young adults; hence, as the basis for an alternative treatment, this study investigated the cytotoxic and antiproliferative effects of naringenin on osteosarcoma cell lines, HOS and U2OS, by using cell counting kit-8 and colony formation assays. DNA fragmentation and the increase in the G2/M phase in HOS and U2OS cells upon treatment with various naringenin concentrations were determined by using the terminal deoxynucleotidyl transferase-mediated dUTP nick-end labeling assay and Annexin V/propidium iodide double staining, respectively. Flow cytometry was performed, and 2′,7′-dichlorodihydrofluorescein diacetate, JC-1, and Fluo-4 AM ester probes were examined for reactive oxygen species (ROS) generation, mitochondrial membrane potential, and intracellular calcium levels, respectively. Caspase activation, cell cycle, cytosolic and mitochondrial, and autophagy-related proteins were determined using western blotting. The results indicated that naringenin significantly inhibited viability and proliferation of osteosarcoma cells in a dose-dependent manner. In addition, naringenin induced cell cycle arrest in osteosarcoma cells by inhibiting cyclin B1 and cyclin-dependent kinase 1 expression and upregulating p21 expression. Furthermore, naringenin significantly inhibited the growth of osteosarcoma cells by increasing the intracellular ROS level. Naringenin induced endoplasmic reticulum (ER) stress-mediated apoptosis through the upregulation of ER stress markers, GRP78 and GRP94. Naringenin caused acidic vesicular organelle formation and increased autophagolysosomes, microtubule-associated protein-light chain 3-II protein levels, and autophagy. The findings suggest that the induction of cell apoptosis, cell cycle arrest, and autophagy by naringenin through mitochondrial dysfunction, ROS production, and ER stress signaling pathways contribute to the antiproliferative effect of naringenin on osteosarcoma cells.

## 1. Introduction

Cancer is a public health problem, with high mortality and disability rates worldwide [1]. According to global cancer statistics, 19.3 million new cancer-related cases and 10 million cancer-related deaths were reported in 2020 [2]. Osteosarcoma is the most prevalent primary bone cancer in children aged 10 to 15 years and young adults [3,4]. The 5-year survival rate of patients with localized osteosarcoma is 65–75% [5]; however, the prognosis of patients with relapse and metastasis is poor, with a 5-year survival rate of 10–20% [6,7]. Currently, treatment for osteosarcoma mainly involves surgical removal, neoadjuvant chemotherapy, and radiation therapy [8]. However, the chemotherapeutic agents currently used cause severe side effects in the majority of patients. Moreover, resistance to chemotherapeutic agents is another challenge in the treatment of osteosarcoma. Hence, new drugs with both low toxicity and high efficacy are urgently required.

Natural products have been widely used as anticancer therapeutics. Sixty percent of drugs approved by the Food and Drug Administration from 1984 to 1994 were isolated from natural sources, especially plants [9,10]. Naringenin [(2*S*)-4′,5,7-trihydroxyflavan-4-one] is present in various herbs and fruits, especially citrus plants [11]. Naringenin possesses antimicrobial, antioxidative, and anticancer properties [12,13,14]. Moreover, naringenin induces cytotoxicity in different types of cancer cells [15]. In 2019, Zhao et al. reported that naringenin (250 μM; 24 h) suppressed the migration of breast cancer cells by arresting the cell cycle at the G0/G1 phase [16]. Furthermore, Song et al. indicated that naringenin (200 μM; 24 h) caused colon cancer apoptosis through p38-dependent ATF3 activation [17]. Naringenin (500 μM; 24 h) enhanced TRAIL-induced apoptosis through the induction of DR5 expression in human A549 cells [18]. However, the effect of naringenin on osteosarcoma remains unclear.

Autophagy, which refers to the intracellular degradation of cytoplasmic materials caused by vacuoles or lysosomes in eukaryotic cells, eliminates and recycles damaged proteins to prolong the lifespan of cells [19]. It is a crucial homeostasis and cell survival mechanism that responds to environmental stresses such as starvation or pathogen infection [20]. Recent accumulating evidence indicates that autophagy also occurs under pathological conditions, such as in neurodegenerative disease or tumor development [21]. Specifically, autophagy is believed to play an important role in tumor development [22]. During the early stages of tumor formation, autophagy functions as a tumor suppressor, and autophagic activity is often impaired in cancer cells. Many anticancer drugs which lead to apoptosis can also induce autophagy-related cell death in cancer cell lines [23]. In osteosarcoma, autophagy is deregulated and functions as a protumoral or antitumoral process to suppress carcinogenesis and support the growth of established tumors [19].

This study investigated the effects of and molecular mechanisms underlying naringenin-induced autophagy and apoptosis and the interaction between autophagy and apoptosis in osteosarcoma cells. The findings of this study can provide the proof-of-concept for evaluating naringenin as an antiosteosarcoma agent.

## 2. Materials and Methods

### 2.1. Material

Primary antibodies for Bak (GTX100063), Bax (GTX109683), Bcl2 (GTX100064), Bcl-xL (GTX105661), GRP78 (GTX113340), GRP94 (GTX103203), PARP (GTX100573), calpain I (GTX102340), calpain II (GTX102499), cytochrome c (GTX108585), beclin1 (GTX134209), p53 (GTX70214), p62 (GTX102361), LC3B (GTX127375), ATG5 (GTX102360) and a voltage-dependent anion channel (VDAC; GTX104745) were purchased from GeneTex International Corporation (Hsinchu City, Taiwan). Caspase-3 (19677-1-AP) and caspase-9 (10380-1-AP) were purchased from Proteintech Group Inc. (Rosemont, IL, USA). CDK1 (MN ABE1403) and cyclin B (MM05373) was purchased from Merck KGaA, Darmstadt, Germany. Anti-mouse and anti-rabbit IgG-conjugated horseradish peroxidase, as well as rabbit polyclonal antibodies specific for β-actin (cat. no. SI-A5441; Sigma-Aldrich; Merck KGaA, Darmstadt, Germany) were used in study. All other chemicals were obtained from Sigma-Aldrich (St. Louis, MO, USA).

### 2.2. Cell Line and Cell Culture

The human osteosarcoma cell lines (U2OS and HOS) and osteoblast cell line (hFOB 1.19) were purchased from the American Type Culture Collection (ATCC; Manassas, VA, USA). The hFOB1.19 osteoblast cells were cultured in a DMEM/F12 medium supplemented with 10% fetal bovine serum (FBS), 2.5 mM L-glutamine, 0.3 mg/mL G418 and 100 units/mL penicillin/streptomycin. Cells were incubated in an atmosphere of 5% CO_2_ at 34 °C and subcultures were changed every 48 h.

The HOS cells were maintained in an Eagle’s Minimum Essential medium supplemented with 10% FBS and 100 units/mL penicillin/streptomycin. The U2OS cells were cultured in a McCoy’s 5A medium supplemented with 10% FBS and 100 units/mL penicillin/streptomycin. (Invitrogen; Thermo Fisher Scientific, Inc., Waltham, MA, USA). Cells were incubated in an atmosphere of 5% CO_2_ at 37 °C and subcultures were changed every 48 h.

### 2.3. Cell Viability/Proliferation Assay

Cells were seeded (6 × 10^3^) in triplicate in a 96-well plate, treated with naringenin at various concentrations (100, 250, and 500 μM), and incubated at 37 °C for 24 h under 5% CO_2_. After incubation, the cell counting kit-8 (CCK-8) assay (10 μL; Sigma-Aldrich, St. Louis, MO, USA) solution was added into each well, and the plate was incubated under the same conditions for 2–5 h. A microplate reader was used to measure absorbance at 450 nm (Bio-Tek Instruments, Winooski, VT, USA).

### 2.4. Colony Formation Assay

HOS and U2OS cells (5 × 10^4^) were seeded in six-well plates and treated with naringenin at the indicated concentrations (100, 250, and 500 μM) in a medium without 10% FBS for 24 h. Untreated cells were used as the control. The culture medium containing 10% FBS was replaced after 24 h of treatment and changed every 2 days without additional naringenin treatment. After incubation for 7 days, visible colonies were washed with PBS, fixed with 4% paraformaldehyde for 15 min, and stained with 0.25% crystal violet for 15 min. The images of colonies were captured through scanning. Subsequently, the plates were washed three times with double-distilled water and then with 33 % (*v*/*v*) acetic acid, and this was followed by the measurement of absorbance at 550 nm. The colony formation assay was repeated three times in duplicate wells. 

### 2.5. Cytosolic and Mitochondrial Protein Extraction

Cytosolic and mitochondrial proteins were extracted from untreated controls and cells treated with naringenin at various concentrations (100, 250, and 500 μM) for 8 h by using the Mitochondria/Cytosol Fractionation Kit (Cat#K256-25; BioVision Inc., Milpitas, CA, USA) according to the manufacturer’s protocol. Cells were collected, washed, and centrifuged for 10 min at 4 °C at 1000 rpm. Cells were then resuspended in the cytosol extraction buffer mix, incubated on ice for 10 min, and repeatedly passed through a 25-gauge needle. The homogenate mix was centrifuged for 10 min at 4 °C at 3000× *g* rpm. The supernatant was collected and centrifuged for 30 min at 4 °C at 15,000× *g* rpm. The supernatant was used as the cytosolic fraction. The pellet was resuspended in a mitochondrial extraction buffer mix and used as the mitochondrial fraction.

### 2.6. Western Blot Analysis

HOS and U2OS cells were treated with naringenin at various concentrations (100, 250, and 500 μM) for 8 h. After treatment, total protein was harvested and lysed in RIPA lysis buffer containing protease inhibitors. Protein concentrations were determined using the bicinchoninic acid assay kit (Sigma-Aldrich, St. Louis, MO, USA). Proteins were separated through 8–15% sodium dodecyl sulfate–polyacrylamide gel electrophoresis and transferred onto polyvinylidene fluoride membranes. The membranes were blocked with a TBST buffer containing 4% BSA for 1 h at room temperature and subsequently incubated with primary antibodies (at a dilution of 1:1000) at 4 °C overnight. The membranes were washed three times with the TBST buffer and then incubated with peroxidase-conjugated secondary antibodies (at a dilution of 1:10,000) for 1 h at room temperature. The blots were visualized using an enhanced chemiluminescence system (EMD Millipore, Billerica, MA, USA) with a UVP BioImaging System (Upland, CA, USA). Each experiment was repeated at least three times. 

### 2.7. DAPI Staining

HOS and U2OS cells were treated with naringenin at various concentrations (100, 250, and 500 μM) for 24 h. After treatment, cells were washed with PBS, fixed in a 3.7% formaldehyde solution for 15 min, permeabilized with 0.1% Triton X-100 for 5 min, and stained with DAPI (1 μg/mL) for 5 min. All samples were examined and photographed using a Nikon Eclipse Ti inverted fluorescence microscope (software version 5.02.01). 

### 2.8. Terminal Deoxynucleotidyl Transferase-Mediated dUTP Nick-End Labeling Assay

Apoptotic cells were quantified using the terminal deoxynucleotidyl transferase-mediated dUTP nick-end labeling (TUNEL) assay, which examines breaks in DNA strands caused during cell apoptosis by using the BD APO-DIRECT kit (BD Biosciences, San Jose, CA, USA; cat. no. 556381). HOS and U2OS cells (2 × 10^6^) were treated with naringenin at various concentrations (100, 250, and 500 μM) for 24 h. After treatment, cells were collected and centrifuged at 950× *g* for 10 min at 4°C. Cells were fixed with 1% paraformaldehyde for 30 min on ice and washed with PBS twice. After the removal of the fixative, 0.5 mL of ethanol was added, and the mixture was incubated at −20 °C for 4 h. Subsequently, ethanol was removed through centrifugation and cellular DNA was obtained. The cellular DNA was stained with TUNEL solution (3 ng/mL TdT enzyme and 0.04 nmol FITC dUTP) at 37 °C for 1 h. After incubation with TUNEL solution, cells were washed with a rinse buffer (1 mL; BD APO-DIRECT kit; BD Biosciences; cat. no. 556381) and centrifuged at 1425× *g* for 10 min at 4 °C. The fluorescein-labeled DNA strand was detected and quantified using the BD Accuri C5 flow cytometer and BD Accuri C6 software (version 1.0.264.21, BD Biosciences).

### 2.9. Annexin V and Propidium Iodide Staining

The Annexin V/propidium iodide (PI) double staining assay was performed to examine cell apoptosis by using the Annexin V/PI detection kit (cat. no. PF00005; Proteintech Group, Inc., Rosemont, IL, USA). HOS and U2OS cells were treated with naringenin at various concentrations (100, 250, and 500 μM) for 24 h. After treatment, cells were collected and washed with PBS twice and then resuspended in a staining buffer containing PI and Annexin V–FITC at room temperature for 30 min; cells were placed in the dark prior to flow cytometry. Cells were analyzed using the BD Accuri C5 flow cytometer and BD Accuri C6 software (version 1.0.264.21, BD Biosciences).

### 2.10. Cell Cycle Analysis Using PI Staining

Apoptotic cells were quantified by examining the cell cycle. HOS and U2OS cells were treated with naringenin at various concentrations (100, 250, and 500 μM) for 24 h and collected through centrifugation (10 min at 950× *g*). Ice-cold ethanol was added to 0.5 mL of cell suspension and the mixture was then incubated at −20 °C for 4 h. Ethanol was removed through centrifugation (15 min at 1425× *g*) and cells were stained with a PI solution (0.1% Triton-X 100, 100 μg/mL of DNase-free RNase A, and 10 μg/mL of PI in PBS). After staining, cells were analyzed using the BD Accuri C5 flow cytometer and BD Accuri C6 software (version 1.0.264.21, BD Biosciences).

### 2.11. Intracellular ROS Production, Ca^2+^ Concentration, and Mitochondrial Mmembrane Potential 

ROS generation was determined using the fluorogenic probe 2′,7′-dichlorodihydrofluorescein (H_2_DCFDA; Thermo Fisher; Waltham, MA, USA). The mitochondrial membrane potential (MMP) was determined using the cationic JC-1 dye (BD Biosciences). The Ca^2+^ concentration was measured using the Ca^2+^-sensitive fluorescent probe Fluo-4 AM (Thermo Fisher). Cells (5 × 10^5^) were plated in six-well plates, grown to confluence, and treated with naringenin at various concentrations (100, 250, and 500 μM) for indicated times. After incubation, cells were stained with H_2_DCFDA (10 μM), Fluo-4 AM (3 μg/mL), and JC-1 (5 μg/mL) to determine ROS production, Ca^2+^ levels, and MMP, respectively. NAC and DPI were used as ROS inhibitors, and BAPTA-AM was used to control the intracellular Ca^2+^ level. Cells were determined using the BD Accuri C5 flow cytometer and BD Accuri C6 software (version 1.0.264.21, BD Biosciences).

### 2.12. Small Interfering RNA Transfection

Small interfering RNAs (siRNAs) against ATG5 (sense: 5′-GUGAGAUAUGGUUUGAAUAdTdT-3′ and antisense: 3′-UAUUCAAACCAUAUCUCACdTdT-5′), Beclin 1 (sense: 5′-GUUUGGAGAUCUUAGAGCAdTdT-3′ and antisense: 3′-UGCUCUAAGAUCUCCAAACdTdT-5′) and a nonspecific scrambled siRNA were purchased from Sigma-Aldrich (Merck KGaA). HOS and U2OS cells (5 × 10^5^) were seeded in six-well plates and transfected with Lipofectamine 3000 (Invitrogen, Rockville, MD, USA) mixed with a serum-free medium containing ATG5 siRNA, Beclin 1 siRNA, and scrambled siRNA. Transfection was performed under 5%CO_2_ at 37 °C for 24 h. 

### 2.13. Autophagy Assay

Cell autophagy was determined using DAPgreen (Dojindo Molecular Technologies, Inc., Kumamoto, Japan) and acridine orange (AO; Sigma-Aldrich; Merck KgaA) according to the manufacturers’ instructions. HOS and U2OS cells (3 × 10^5^) were seeded in six-well plates overnight. Subsequently, cells were washed with PBS and incubated with DAPgreen and AO for autophagy detection at 37 °C for 30 min. After cells were washed twice with PBS, they were treated with the indicated concentrations of naringenin (100, 250, and 500 μM) for 24 h. After treatment, the image of the cells was obtained under a 200× microscope to determine the visual intensity of green fluorescence (DAPgreen) and orange/green fluorescence (AO; autophagy) using a Nikon ECLIPSE Ti and NIS-Elements AR microscope (software version 5.02.01).

### 2.14. Caspase Activity Assay

The caspase activity assay is based on the ability of active enzymes to cleave chromophores from the enzyme substrate Ac-DEVD-pNA (caspase-3; cat. #1008) or Ac-LEHD-pNA (caspase-9; cat. #1076; BioVision Inc., Milpitas, CA, USA). To examine the activity of caspase-3 and -9, HOS and U2OS cell lysates were prepared and incubated with caspase-3 and -9 substrates. Immunocomplexes were incubated with the peptide substrate in the assay buffer (100 mM NaCl, 50 mM 4-(2-hydroxyethyl)-1-piperazine ethanesulfonic acid, 10 mM dithiothreitol, 1 mM EDTA, 10% glycerol, and 0.1% CHAPS; pH 7.4) at 37 °C for 6 h. The release of *p*-nitroaniline was monitored using an enzyme-linked immunosorbent assay reader at 405 nm. The results are presented as the percentage change in activity compared with the untreated control.

### 2.15. Transmission Electron Microscopy

HOS and U2OS cells (1 × 10^5^) were treated with naringenin as previously described. Cells were fixed for 10 min in 50% Karnovsky’s fixative. Cells were collected and centrifuged at 1500× *g* for 5 min. The pellet was washed and stored in 70% Karnovsky’s fixative at 4 °C until embedding and then analyzed by means of a transmission electron microscope. The sections were observed under a JEOL JEM-1400 electron microscope (Tokyo, Japan).

### 2.16. Statistical Analysis

All results were analyzed using the GraphPad Prism program (GraphPad, San Diego, CA, USA). Quantified results were expressed as the mean ± standard deviation (SD) and analyzed with one-way ANOVA followed by Fisher’s least significant difference (LSD) post-hoc test. For all results, *p* < 0.05 was considered a significant difference.

## 3. Results

### 3.1. Induction of Cell Apoptosis in Human Osteosarcoma Cells by Naringenin

Naringenin contains the skeleton structure of flavanone and three hydroxy groups at 4′, 5′, and 7′ carbons ((2*S*)-4′,5′,7′-trihydroxyflavan-4-one; Figure 1A). To determine the effect of naringenin on human osteosarcoma, HOS and U2OS cells were treated with varying concentrations of naringenin for 24 h. The results indicated that naringenin reduced the viability of osteosarcoma cells but not normal hFOB 1.19 cells in a concentration-dependent manner (Figure 1B–D), indicating that naringenin selectively inhibited the growth of osteosarcoma cells but exhibited less cytotoxicity in normal human bone cells. In addition, the antiproliferative effect of naringenin on HOS and U2OS cells was evaluated through a colony formation assay. The results demonstrated that compared with no treatment, naringenin treatment significantly reduced the number of colonies in a dose-dependent manner (Figure 1E,F). Cell apoptosis caused by naringenin was examined using with DAPI staining. The results indicated that naringenin increased DNA condensation and morphological changes in HOS and U2OS cells (Figure 1G). 

To explore whether naringenin inhibited the viability and proliferation of HOS and U2OS cells by inducing apoptosis, apoptotic cells were detected by performing Annexin V/PI double-labeling and the TUNEL assay. Treatment with varying concentrations of naringenin significantly promoted the apoptosis of osteosarcoma cells in a dose-dependent manner (Figure 2A,B). Similar results were obtained in the TUNEL assay (Figure 2C,D). To investigate the inhibitory effects of naringenin on the proliferation of HOS and U2OS cells, we examined cell cycle distribution after 24 h treatment with naringenin. The percentages of naringenin-treated osteosarcoma cells in the G2/M phase were significantly higher than those of control cells (Figure 2E). To verify this change, the levels of G2/M phase regulatory proteins (CDK1 and cyclin B) were examined by western blotting. The results indicated that compared with control cells, naringenin-treated cells exhibited decreased CDK1 and cyclin B levels and upregulated p21 expression (Figure 2F). These findings indicated that naringenin induced cell apoptosis and prolonged G2/M arrest in human osteosarcoma cells.

### 3.2. Induction of ROS-Mediated ER Stress by Naringenin in Osteosarcoma Cells

ROS plays a crucial role in the apoptotic process in many cell types [24]. ER stress is induced by the accumulation of ROS, leading to mitochondrial dysfunction and apoptosis [25]. As shown in Figure 3A,B, naringenin induced ROS production and accumulation. Pretreating cells with the antioxidant NAC and the inhibitor of the flavoprotein-dependent oxidase DPI reduced ROS production and naringenin-induced cell apoptosis in osteosarcoma cells (Figure 3C,D). 

To determine whether naringenin induced apoptosis by triggering ER stress, we first examined its effect on the mobilization of Ca^2+^. The Ca^2+^ level was significantly increased in naringenin-treated cells compared with control cells (Figure 4A,B). The results revealed that aberrant Ca^2+^ imbalance increased in naringenin-treated cells in a dose-dependent manner. We next determined whether ER stress-associated proteins, glucose-regulated proteins (GRP78 and GRP 94), and calpain proteins (calpain I and calpain II) are induced by naringenin in osteosarcoma cells. As shown in Figure 4C, naringenin increased the expression of GRP78, GRP94, and calpain I and II in a dose-dependent manner. Moreover, cells pretreated with BAPTA-AM, a Ca^2+^ cell-permeable chelator, markedly reduced naringenin-mediated cell apoptosis (Figure 4D). In addition, naringenin-induced Ca^2+^ release was inhibited by ROS inhibitors (DPI or NAC; Figure 4E). These findings indicated that naringenin induced cell apoptosis in osteosarcoma cells through ROS production and caused ER stress.

### 3.3. Involement of Cell Apoptosis in Naringenin-Induced Mitochondrial Dysfunction in Human Osteosarcoma 

ROS production and accumulation in mitochondria reduce MMP, thereby triggering the mitochondrial apoptotic pathway [26]. To examine the effect of naringenin on HOS and U2OS cells, JC-1 staining was performed to determine the fluorescence ratio of orange and green fluorescence between normal and unhealthy mitochondria. Orange fluorescence disappeared in cells when observed under the 200× microscope, indicating mitochondrial dysfunction (Figure 5A). To verify the loss of MMP, HOS and U2OS cells were analyzed using with flow cytometry, and the results revealed a shift in MMP in naringenin-treated cells (Figure 5B,C). The findings indicated the loss of MMP in cells treated with various naringenin concentrations. Mitochondrial and cytosolic proteins, cytochrome c, and proapoptotic/antiapoptotic proteins were examined in cells treated with naringenin. The release of cytochrome c due to mitochondrial dysfunction was upregulated by protein–protein interactions between Bcl-2 proteins (Figure 5D,E). Proapoptotic proteins, Bak and Bax, and cytochrome c released in the cytosol were upregulated with the increasing concentration of naringenin (Figure 5D,E). In the mitochondrial pathway of apoptosis, the downstream signaling of caspases was activated after treatment with naringenin. To determine the primary mediators of apoptosis, caspase-3 and caspase-9 were examined in naringenin-treated HOS and U2OS cells. The results revealed the upregulation of caspases, including that of the PARP cleavage (Figure 5F). To determine the intrinsic pathway of naringenin-induced apoptosis, caspase-3 and caspase-9 activities were examined separately (Figure 5G,H). The activities of both caspases were associated with naringenin-induced cell apoptosis. Pretreatment of cells with a caspase-3 inhibitor (z-DEVD-FMK) or caspase-9 inhibitor (z-LEHD-FMK) inhibited naringenin-induced cell apoptosis (Figure 5I). Thus, these data demonstrated that naringenin induced mitochondrial dysfunction and the subsequent release of cytochrome c and activation of caspases-9 and caspases-3 in human osteosarcoma cells. 

### 3.4. Triggering of Autophagy in Human Osteosarcoma Cells by Naringenin

Because cell autophagy regulates cell death, we examined whether naringenin can induce autophagy. The autophagy phenomenon was observed in HOS and U2OS cells treated with naringenin. Using a transmission electronic microscope, it was observed that naringenin induced the formation of numerous intracytoplasmic vacuoles in HOS cells (Figure 6A). In addition, to observe naringenin-induced autophagy, HOS and U2OS cells were stained with DAPgreen, and the green fluorescence intensity within cells was detected. The intensity increased with the concentration of naringenin (Figure 6B). Moreover, AO staining revealed the accumulation of acidic vesicles in HOS and U2OS cells (Figure 6C). Furthermore, we examined the expression of several proteins that serve as markers of autophagy. Naringenin increased the level of microtubule-associated protein light chain 3 (LC3)-II and the expression of ATG5, Beclin 1, and p62 in a dose-dependent manner (Figure 6D). To determine the role of autophagy in naringenin-induced cell death, siRNAs of Beclin 1 and ATG5 were transfected in HOS and U2OS cells. The results of the CCK-8 assay revealed that Beclin 1 and ATG5 siRNAs suppressed the naringenin-induced loss of cell viability (Figure 6E), indicating that the autophagic mechanism of ATG5 and Beclin 1 involved in naringenin-induced cell death promotes either survival or death and is associated with signaling transduction in programmed cell apoptosis. Furthermore, we examined naringenin-induced cell autophagy by using ROS inhibitors. We performed DAPgreen staining to observe the green fluorescence intensity (Figure 6F). Cell images were obtained using a fluorescence microscope, and it was observed that after adding the inhibitors, the autophagosome (white arrow) significantly decreased in naringenin-treated cells compared with control cells. The results indicate that naringenin-induced intracellular ROS production triggered autophagic signaling in human osteosarcoma cells. 

## 4. Discussion

Grapefruit is important not only as a fruit but also in traditional medicine [27]. It is a rich source of bioactive compounds that may serve as chemopreventive agents in cancer therapy [28]. Naringenin is the active component in grapefruit extract [29]. Naringenin exhibits various bioactivities including antioxidative, anti-inflammatory, and anticancer effects. Naringenin exerts in vitro effects on various cancer cell lines at high concentration; eg., breast cancer (SKBR3 and MDA-MD-231; 250 μM) [16,17,18,30,31] and liver cancer cell lines (HepG2; 100–200 μM) [31]. In this study, we found that the IC_50_ values for HOS and U2OS osteosarcoma cells after 24 h treatment with naringenin were 276 and 389 μM, respectively. Although a high naringenin concentration was used during the study, naringenin selectively inhibited the growth of osteosarcoma cells and exhibited less cytotoxicity in normal human bone cells. This is the first study to report the cytotoxic and antiproliferative effects of naringenin on human osteosarcoma HOS and U2OS cells. In our study, the results of CCK-8 and colony formation assays and DAPI staining revealed that naringenin significantly inhibited the proliferation of HOS and U2OS cells. Moreover, naringenin induced caspase-dependent apoptosis in HOS and U2OS cells, as determined through flow cytometry, TUNEL staining, and the expression of apoptosis-related proteins. In this study, we observed that naringenin exhibited potent antitumor activity against osteosarcoma in vitro. Naringenin-induced ROS overproduction resulted in ER stress activation and autophagy, which caused cell apoptosis. The findings indicated that naringenin plays multiple roles in signaling pathways in human osteosarcoma cells.

Cell apoptosis and autophagy, which are involved in the regulation of cancer cell death, have been widely studied. Autophagy plays a critical role in maintaining homeostasis and the pathogenesis of many human diseases, including cancer [32]. Autophagy, a dynamic and conserved catabolic process, plays dual roles in cell survival and death [20]. In some circumstances, autophagy can lead to cell death either in collaboration with apoptosis or as a backup mechanism when apoptosis is defective [33]. Autophagy can be induced through acute exposure to resveratrol while prolonging the activation of the caspase-mediated cell death pathway [34]. Because Bcl-2 family proteins control the release of cytochrome c during mitochondrial dysfunction, Beclin 1 can be upregulated to activate the autophagy pathway with autophagy-related genes (ATG) and their protein products [35]. Cleavage of LC3 is an essential step in autophagosome formation [36]. Thus, in this study, we examined whether naringenin induces autophagy in osteosarcoma cells. We evaluated the expression levels of the autophagy markers LC3-I/LC3-II, Beclin 1, and p62. Transmission electron microscopy was mainly used for the detection of autophagosomes and the autophagy level. Our results indicated that naringenin induced autophagy in osteosarcoma cells.

Programmed cell apoptosis can be triggered by certain endogenous or exogenous signals through various ER stress-induced signals. Condensation of chromatin, fragmentation of DNA, and shedding of small fragments from cells are the innate cellular responses of apoptosis [37]. We examined mitochondrial dysfunction-induced apoptosis and observed that caspases play critical roles in the apoptotic cascade [38]. Our results revealed that naringenin increased the protein expression of caspase-3 and caspase-9. Moreover, Bcl-2 family proteins are mitochondrial apoptotic regulators that can be either proapoptotic proteins, such as Bax, or antiapoptotic proteins, such as Bcl-2. Naringenin upregulated Bax and Bak protein expression and downregulated Bcl-2 protein expression. Our results indicated that naringenin induced apoptosis through caspase-dependent apoptotic pathways by activating caspase-3/-9 and altering Bax/Bcl-2 in HOS and U2OS cells.

ROS acts as a second messenger in cell signaling by regulating various biological processes in normal and cancer cells [39]. The ROS production is shared by most chemotherapeutics due to its role in triggering cell death [40,41]. Excessive ROS generation can result in oxidative stress, DNA damage, and cell death through apoptosis [42]. For example, Zhang et al. [43] reported that libertellenone H treatment significantly increased intracellular ROS generation and induced apoptosis in human pancreatic cancer cells in a ROS-dependent manner. Similarly, curcumin derivatives markedly increased ROS production in colon cancer cells [44]. Furthermore, naringenin-inhibited cell growth was effectively increased through intercellular ROS production [45]. ROS is crucial in cellular apoptosis and autophagy pathways [46]. For example, Zhang et al. [47] demonstrated that the induction of autophagy increased ROS-mediated apoptosis in human bladder cancer cells. They reported that apoptosis and autophagy were dependent on ROS production. The obtained results suggest that ROS plays a vital role in cancer cell apoptosis and autophagy.

Accumulating evidence has indicated that apoptosis is regulated by ER stress [48,49]. The ER is a major organelle in eukaryotic cells that is involved in protein processing, intracellular calcium storage, and lipid synthesis. In the present study, naringenin exerted potent cytotoxic and apoptotic effects on human osteosarcoma cells. We found that the intracellular ROS level was increased after naringenin treatment and that the antioxidant NAC rescued the apoptosis level. In addition, naringenin stimulated the expression of ER stress-related proteins including GRP78, GRP94, calpain I, and calpain II. Naringenin-induced apoptosis was suppressed by the ER stress inhibitor BAPTA-AM in HOS and U2OS cells. These results suggest that apoptosis is regulated by ER stress signaling. Moreover, ROS inhibitors (DPI and NAC) suppressed calcium release and autophagosome formation. Therefore, naringenin induced ER stress and autophagy by promoting ROS generation, thus resulting in cellular apoptosis.

In conclusion, we demonstrated that naringenin induced apoptosis through the ROS-mediated ER stress signaling pathway and autophagy. Although a high concentration of naringenin was required in vitro to cause cancer cell death, naringenin is a potential agent against human osteosarcoma cells. Future studies examining novel naringenin-based derivatives are warranted.

## Figures and Tables

**Figure 1 molecules-27-00373-f001:**
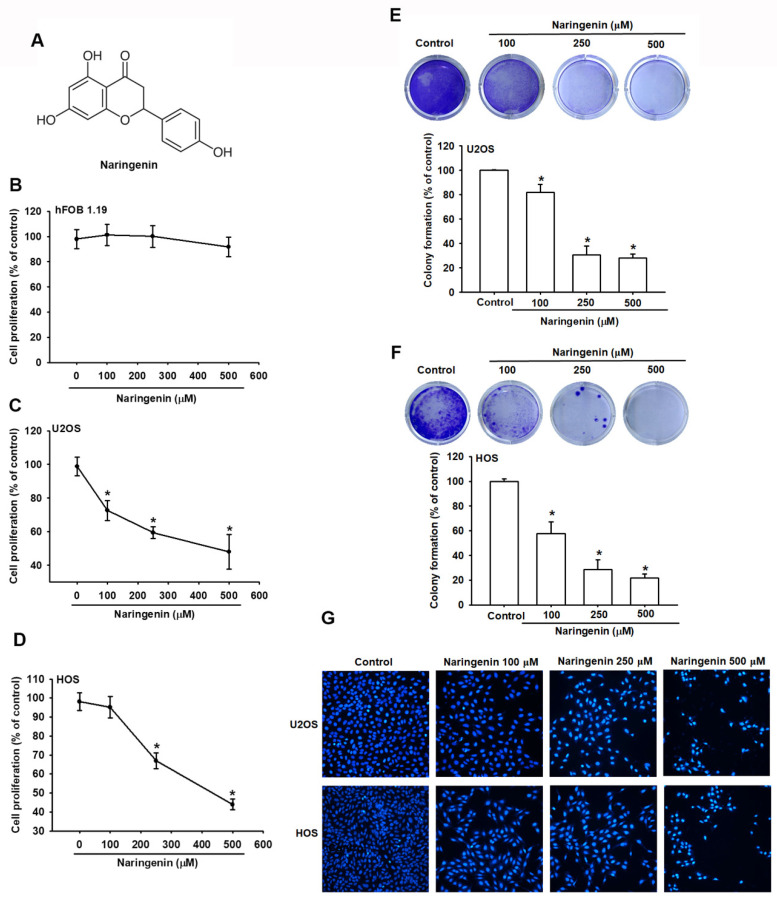
Naringenin inhibits the proliferation of human osteosarcoma cells. (**A**) Structure of naringenin. (**B**–**D**) hFOB 1.19, HOS, and U2OS cells were treated with the indicated concentration of naringenin for 24 h, and cell proliferation was determined using the CCK-8 assay. (**E**,**F**) Colony formation assay was performed in HOS and U2OS cells treated with naringenin at various concentrations (100, 250, and 500 μM) for 7 days. Quantitative results indicated the number of colonies per group. (**G**) After cells were incubated with various concentrations of naringenin for 24 h, the nucleus morphology was determined through 4′,6-diamidino-2-phenylindole staining, and cells were photographed. Magnification, ×200. Results are expressed as the mean ± SD of four independent experiments. * *p* < 0.05 compared with control.

**Figure 2 molecules-27-00373-f002:**
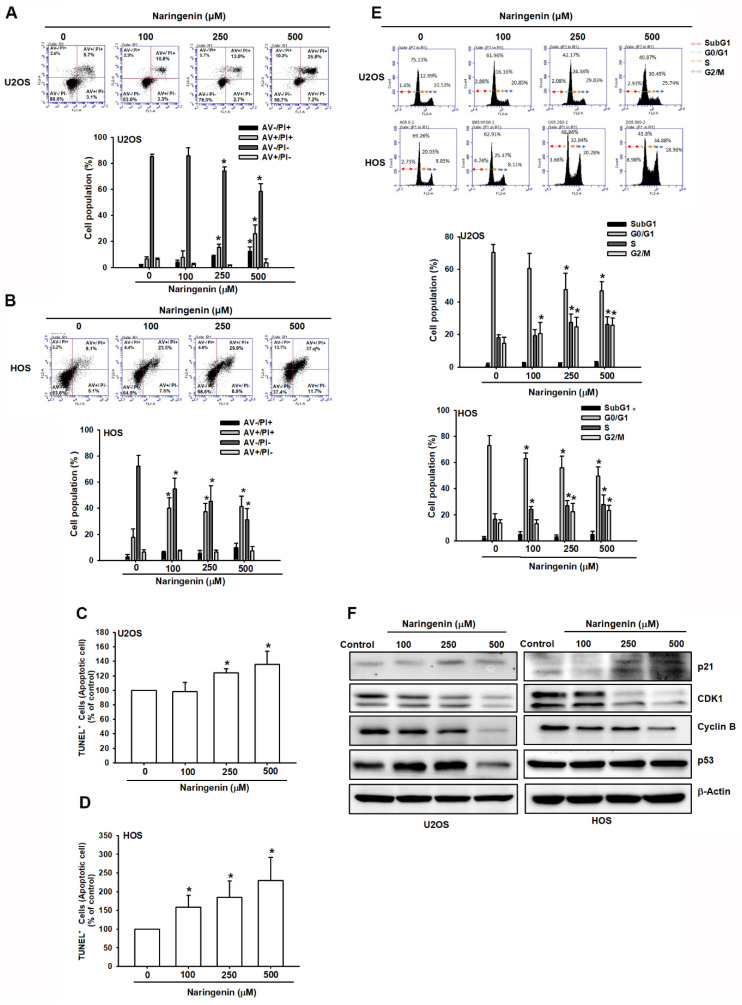
Naringenin induced apoptosis and cell cycle arrest at the G2/M phase in human osteosarcoma cells. (**A**–**D**) Cells were treated with naringenin at various concentrations (100, 250, and 500 μM) for 24 h. Apoptotic cells were determined using with Annexin V/PI and TUNEL staining. (**E**) Flow cytometry was performed to determine the cell cycle distribution of HOS and U2OS cells treated with naringenin at various concentrations (100, 250, and 500 μM) for 8 h. (**F**) Levels of p21, CDK1, cyclin B, and p53 proteins in naringenin-treated cells for 8 h were detected through western blotting. Results are expressed as the mean ± SD of four independent experiments. * *p* < 0.05 compared with control.

**Figure 3 molecules-27-00373-f003:**
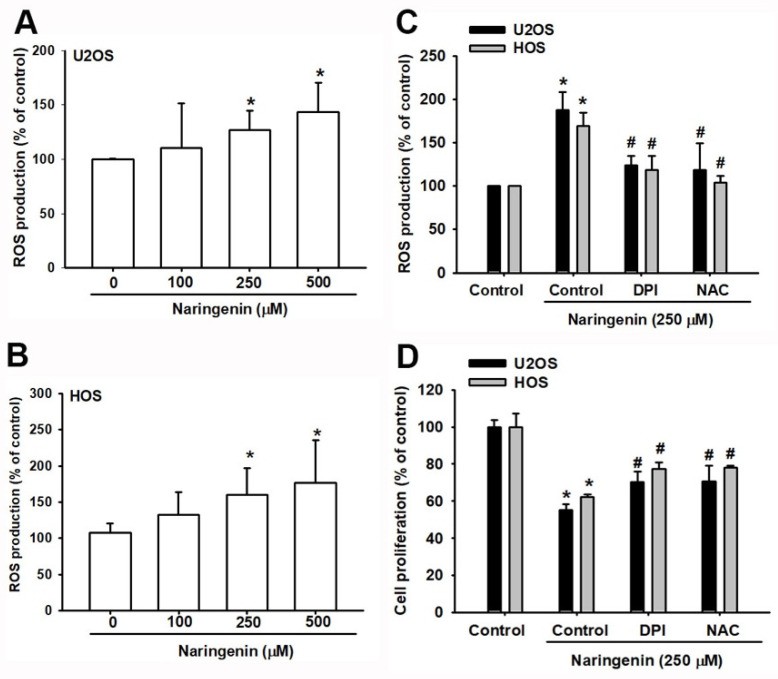
Naringenin induces reactive oxygen species (ROS) generation in human osteosarcoma cells. (**A**,**B**) Cells were treated with naringenin (100, 250, and 500 μM) for 1 h and stained with H_2_DCFDA. The percentage of ROS production was determined by flow cytometry. (**C**) Cells were treated as described in A-B. ROS inhibitors inhibited ROS production. (**D**) After cells were pretreated with a ROS inhibitor for 1 h, followed by stimulation with naringenin for 24 h, cell viability was examined by using the CCK-8 assay. Results are expressed as the mean ± SD of three independent experiments. * *p* < 0.05 compared with the control group; # *p* < 0.05 compared with the naringenin-treated group.

**Figure 4 molecules-27-00373-f004:**
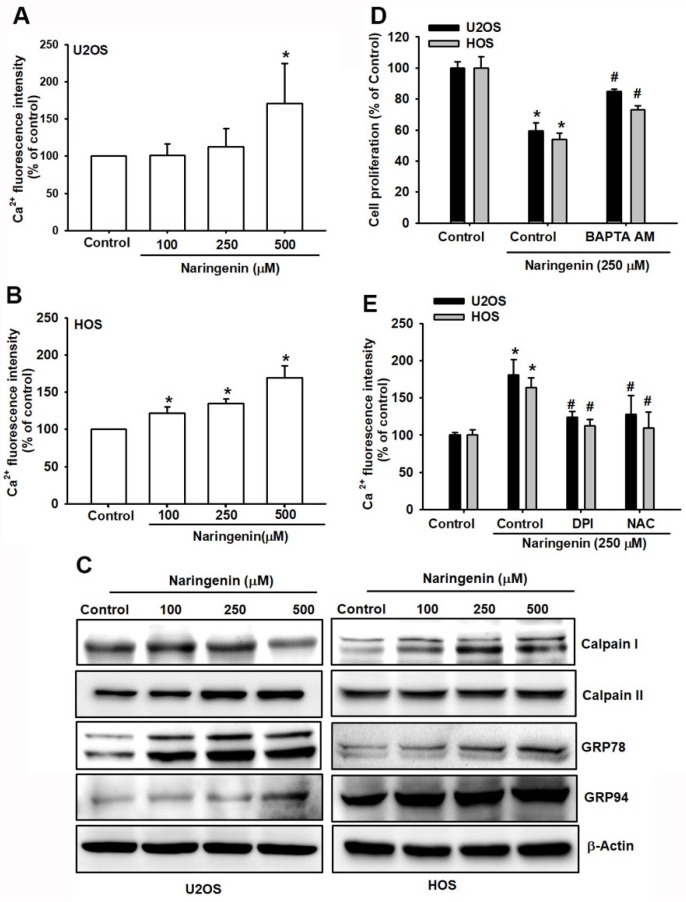
Naringenin triggers Ca^2+^ release and causes endoplasmic reticulum (ER) stress in human osteosarcoma cells. (**A**,**B**) Cells were treated with naringenin (100, 250 and 500 μM) for 5 h and stained with Fluo-4 AM. Ca^2+^ expression was examined by using flow cytometry. (**C**) After cells were treated with varying concentrations of naringenin for 8 h, the protein expression of GRP78, GRP94, calpain I, and calpain II was examined using with western blotting, and β-actin was used as the internal control. (**D**) After cells were pretreated with BAPTA-AM for 1 h, followed by stimulation with naringenin for 24 h, cell viability was examined using the CCK-8 assay. (**E**) After cells were pretreated with a ROS inhibitor for 1 h, followed by stimulation with naringenin for 5 h, Ca^2+^ expression was examined by using flow cytometry. Results are expressed as the mean ± SD of four independent experiments. * *p* < 0.05 compared with the control group. # *p* < 0.05 compared with the naringenin-treated group.

**Figure 5 molecules-27-00373-f005:**
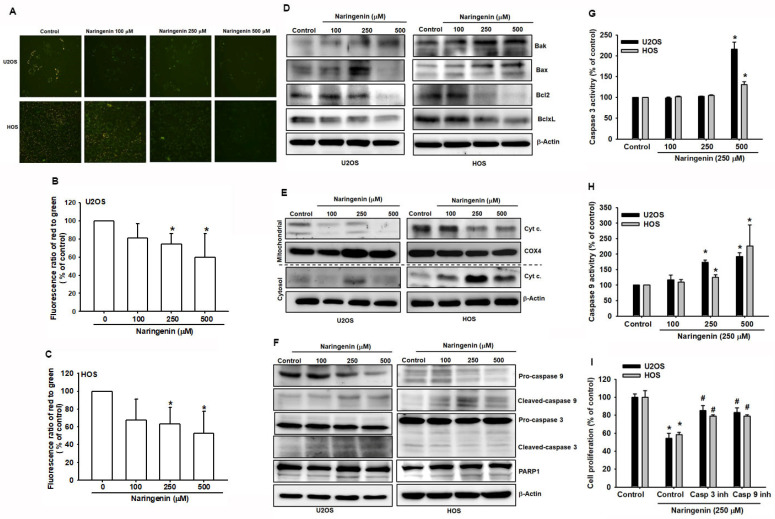
Naringenin triggers endoplasmic reticulum (ER) stress-related mitochondrial dysfunction in human osteosarcoma cells. (**A**–**C**) Cells were treated with naringenin at various concentrations (100, 250, and 500 μM) for 24 h and stained with JC-1 to determine the reduction in mitochondrial membrane potential. The percentage of cells was analyzed by using flow cytometry, and images were obtained using a fluorescence microscope. Magnification, ×200. (**D**– **F**) Cells were treated as described in A–B. The expression of mitochondrial dysfunction-related proteins, mitochondrial and cytosolic cytochrome c, Bax, Bak, Bcl-2, Bcl-xl, caspase-3, caspase-9, and PARP was determined by western blotting. (**G**,**H**) Cells were treated as described in A–B. The upregulation of caspase-3 and caspase-9 activities was determined and analyzed using the caspase activity assay. (**I**) Cells were pretreated with caspase inhibitors separately, followed by naringenin treatment (250 μM), and cell proliferation was determined using the CCK-8 assay. Results are expressed as the mean ± SD of four independent experiments. * *p* < 0.05 compared with the control group. # *p* < 0.05 compared with the naringenin-treated group.

**Figure 6 molecules-27-00373-f006:**
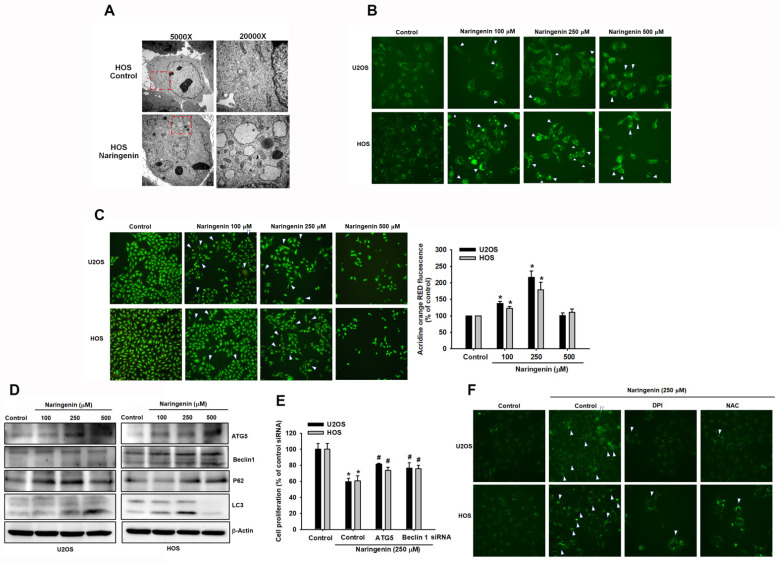
Naringenin initiates autophagy signaling responses in human osteosarcoma cells. (**A**) Electron microscope images of HOS cells were obtained with and without naringenin treatment for 8 h. (**B**,**C**) Cells were treated with naringenin for 8 h and stained with DAPgreen and acridine orange to determine the induction of autophagosome. (**D**) The expression of ATG5, Beclin 1, p62, and LC3-II proteins was determined by western blotting after naringenin treatment for 8 h. (**E**) siRNAs of ATG5 and Beclin 1 were transfected in cells for 24 h, followed by treatment with naringenin for 24 h. Cell proliferation was determined using the CCK-8 assay. (**F**) Cells were pretreated with ROS inhibitors, followed by naringenin treatment (250 μM) for 8 h, and were then stained with DAPgreen to determine autophagosome formation. Images were obtained using a fluorescence microscope. Magnification, ×200. Results are expressed as the mean ± SD of four independent experiments. * *p* < 0.05 compared with the control group. # *p* < 0.05 compared with the naringenin-treated group.

## Data Availability

All data and materials are available for verification as needed.

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
