# Peer review of "Naringenin Induces ROS-Mediated ER Stress, Autophagy, and Apoptosis in Human Osteosarcoma Cell Lines"

_molecules, 2022, doi:10.3390/molecules27020373_

Round 1
Reviewer 1 Report
The authors described that a natural product naringenin induces generation of reactive oxygen species mediated endoplasmic reticulum stress, autophagy and apoptosis in human osteosarcoma cell lines. The data in the article are very detailed, but there are points need to be improved. Some of my comments are as follows.
- Most importantly, the compound was used in such high concentrations and showed modest efficacy that there was no need to study it. In other words, this article should give a detailed explanation of the necessity and reasons for this research, so as to enhance readers' interest.
- What is the logical relationship between generation of ROS, endoplasmic reticulum stress, autophagy and apoptosis induced by naringenin? The author should design some experiments to clarify, otherwise the article will just be a list of some data.
- In figure 1G, the morphology of cell chromatin condensation was detected by DAPI staining under fluorescence microscope, but there is no chromatin condensation at all as I see it. It can only be concluded that the number of some cells decreased after the drug treatment. This is also strongly associated with subsequent apoptosis, which is worth further confirmation if this result is inconsistent with the description.
Author Response
Dear Molecules Editorial,
Thank you for your positive response regarding the publication of our manuscript in Molecules. We are glad that the reviewers find the results of this study useful to the field and greatly appreciate their valuable and helpful suggestions.
We have prepared a new revised version of the manuscript where we take into consideration the suggestions of the academic editor and reviewers. The manuscript has been carefully revised to address the suggested aspects and a point-by-point answer to the referee’s comments follow below. We used highlight (red font) to mark the changes in Word file.
We hope that these changes will be sufficient to make our manuscript suitable for publication in Molecules.
Reviewers' comments:
Reviewer 1
Q1. Most importantly, the compound was used in such high concentrations and showed modest efficacy that there was no need to study it. In other words, this article should give a detailed explanation of the necessity and reasons for this research, so as to enhance readers' interest.
Ans: The purpose of naringenin been studied was to determine the basic mechanism and function in osteosarcoma condition. In order to maximize the efficiency, modifying and synthesizing derived-compound is the next step to analyze based on this natural compound structure and the interaction between cancers. Description was written from line 442 to 443.
Q2. What is the logical relationship between generation of ROS, endoplasmic reticulum stress, autophagy and apoptosis induced by naringenin? The author should design some experiments to clarify, otherwise the article will just be a list of some data.
Ans: Evidences and studies have suggested that flavonoids have the function of anticancer properties through various mechanism including cellular proliferation inhibition, the induction of apoptosis, autophagy, necrosis, cell cycle arrest, senescence, the impairment of cell migration, invasion, tumor angiogenesis, and the reduction of multidrug resistance in tumor cells. Flavonoids, a group of natural polyphenolic compounds characterized by multiple targets participating in multiple pathways, have been widely studied in different types of cancer to promote as an alternative treatment.
In this study, we found that autophagy phenomena and apoptosis have shown with the naringenin treatment in osteosarcoma cells. However, flavonoid-induced autophagy commonly interacts with other mechanisms, comprehensively influencing the anticancer effect. Accordingly, targeted autophagy may become the core mechanism of flavonoids in the treatment of tumors. This paper founds the flavonoid-induced autophagy of tumor cells and their interaction with other mechanisms (ROS, endoplasmic reticulum stress and apoptosis), so as to provide a comprehensive and in-depth account on how flavonoids exert tumor-suppressive effects through autophagy.
Q3. In figure 1G, the morphology of cell chromatin condensation was detected by DAPI staining under fluorescence microscope, but there is no chromatin condensation at all as I see it. It can only be concluded that the number of some cells decreased after the drug treatment. This is also strongly associated with subsequent apoptosis, which is worth further confirmation if this result is inconsistent with the description.
Ans: We thank the reviewer for your comments. The data have been checked and updated in Figure 1G for better observation for readers, the intensity fluorescence. After the naringenin treatment, apoptosis result is observed in HOS and U2OS subsequently. With DAPI staining, DNA condensation illustrates the result of apoptosis which is caused through DNA condensation formation leading to cell death.
Reviewer 2 Report
Running title: Naringenin induces generation of reactive oxygen species mediated endoplasmic reticulum stress, autophagy and apoptosis 3 in human osteosarcoma
The authors highlight the potential use of naringenin as an inducing agent of apoptosis mediated by autophagy, intracellular ROS production and ER stress in Osteosarcoma. It is an interesting and novel natural alternative treatment. However, certain major/minors revisions are required, which are discussed below before considering its publication.
Some of my concerns/suggestions are:
- Naringenin is a flavonoid, which are known molecules with antioxidant properties in the cell. How is it possible that Naringenin induces ROS in this case? Has any possible mechanism been described by which this occurs?
- Page 2, line 76: Autophagy is much more than "an intracellular degradation", please redact that sentence better, also recognizing the highly conserved role of autophagy among all eukaryotic organisms.
- Page 2, line 77-79: I consider it relevant that the authors highlight the role of autophagy as a mechanism of acquisition of resistance to chemotherapy in several cancer models.
- 1. Material: Because in general there are several references for the same marker, and because of their importance for replicating these results in the future, I consider it necessary that the authors provide each of the references for the antibodies used, either in this chapter or as a supplementary table.
- 2. Cell Culture: Which basal media was used to each cell line? The three media were supplementary with the same supplements? It is no clear.
- 3. Cell Viability/Proliferation Assay: What among of CCK-8 solution was added into each well?
- 4. Colony Formation Assay (and Figure 1 E-F): The culture media which were changed every 2 days kept fresh naringenin? The media with drug was applied since the firth day even when few cells were present in the plates?
- 5. Cytosolic and Mitochondrial Protein Extraction: Give the name of the commercial protein assay.
- 6. Western Blot Analysis: All primary antibodies were used at 1:1,000 for 1 h at room temperature?
- 12. Intracellular ROS activity… line 188-190: Which was the incubation time of each staining?
- 13. Small-Interfering RNA (siRNA) Transfection: All Small-interfering RNA sequences should be given
- Result
- Line 240: “The results were demonstrated the antitumor formation effect of naringenin in osteosarcoma cells” should be correctly written. May be “The results demonstrated the antitumor effect of naringenin on osteosarcoma cells”
- Figure 1 E-F: What happen with the normal cells? There were effects on them? Could the authors show that results? In general, these results are lost all over the figures.
- Figure 2 A-B: In panel A the legend is incomplete. The meaning of AV and PI should be defined in the caption of the figure and all the plots showing the indicated % of cells in each quadrant should be included in an additional supplementary figure.
- Figure 2E: I not able to see any difference among control and treatment conditions. Could the authors explain better that differences if they really exist.
- Figure 2 F-H (in fact in the caption say F, G; H is interchanged with I): could not see anything in panel F, please indicate correctly all the information.
- Figure 6, line 385: for 8h, no “or 8h”. “The ratio of orange and green fluorescence within cells” should be graphically represented as possible, contrary that phrase (line 360-361) should be changed.
Author Response
Q1. Naringenin is a flavonoid, which are known molecules with antioxidant properties in the cell. How is it possible that Naringenin induces ROS in this case? Has any possible mechanism been described by which this occurs?
Ans:
Glutathione (GSH) is an important intracellular antioxidant and acts as a regulator of cellular redox state protecting cells from damage caused by lipid peroxides, reactive oxygen and nitrogen species, and xenobiotics. Recent studies have highlighted the importance of GSH in key signal transduction reactions as a controller of cell differentiation, proliferation, apoptosis, ferroptosis and immune function. Molecular changes in the GSH antioxidant system and disturbances in GSH homeostasis have been implicated in tumor initiation, progression, and treatment response. Hence, GSH has both protective and pathogenic roles. Although in healthy cells, it is crucial for the removal and detoxification of carcinogens that elevated GSH levels in tumor cells associating with tumor progression, increased metastasis, and increase resistance to chemotherapeutic drugs. In contrast, reduction of GSH antioxidant system in tumor becomes a mean for increased respond for cell death signaling or decreased drug resistance.
According to study, naringenin is a natural polyphenolic compound, which containing phenol ring-containing polyphenolics only catalytic amounts of H2O2 and phenols were required for GSH or NADH co-oxidation. This was attributed to phenoxyl radicals because oxygen activation (resulting in superoxide radical and H2O2 formation) accompanied the GSH or NADH co-oxidation; therefore, naringenin can cause GSH to be depleted and increase ROS activity for inducing tumor cell apoptosis (1, 2).
Q2. Page 2, line 76: Autophagy is much more than "an intracellular degradation", please redact that sentence better, also recognizing the highly conserved role of autophagy among all eukaryotic organisms.
Ans: Line 76 has been re-phased for better description from line77 to 84.
Q3. Page 2, line 77-79: I consider it relevant that the authors highlight the role of autophagy as a mechanism of acquisition of resistance to chemotherapy in several cancer models.
Ans: The role of autophagy and resistance of chemotherapy have been described in line 81 to 84.
Q4. Material: Because in general there are several references for the same marker, and because of their importance for replicating these results in the future, I consider it necessary that the authors provide each of the references for the antibodies used, either in this chapter or as a supplementary table.
Ans: We thank the reviewer for your comments. Listed antibodies have been included for its reference numbers in material part (Line 98-104).
Q5. Cell Culture: Which basal media was used to each cell line? The three media were supplementary with the same supplements? It is no clear.
Ans: We thank the reviewer for your comments. The data have been provided in Medium for proper cell lines has been included in line 113 and 115, and same supplements were added.
Q6. Cell Viability/Proliferation Assay: What among of CCK-8 solution was added into each well?
Ans: The volume of CCK-8 used has been provided in line 122.
Q7. Colony Formation Assay (and Figure 1 E-F): The culture media which were changed every 2 days kept fresh naringenin? The media with drug was applied since the firth day even when few cells were present in the plates?
Ans: Description of colony formation assay has been updated to clarify the misunderstanding from line 127-129
Q8. Cytosolic and Mitochondrial Protein Extraction: Give the name of the commercial protein assay.
Ans: We thank the reviewer for your comments. The description has been provided in Method from line 137-146
Q9. Western Blot Analysis: All primary antibodies were used at 1:1,000 for 1 h at room temperature?
Ans: The incubation time for primary antibodies has been updated in line 153.
Q10. Intracellular ROS activity… line 188-190: Which was the incubation time of each staining?
Ans: Time of each incubation has been included in line 203.
Q11. Small-Interfering RNA (siRNA) Transfection: All Small-interfering RNA sequences should be given
Ans: The siRNA sequences of ATG5 and Beclin1 have been included from line 205-208.
Q12. Line 240: “The results were demonstrated the antitumor formation effect of naringenin in osteosarcoma cells” should be correctly written. May be “The results demonstrated the antitumor effect of naringenin on osteosarcoma cells”
Ans: Line 240 has been updated to reviewer’s suggestion at line 252.
Q13. Figure 1 E-F: What happen with the normal cells? There were effects on them? Could the authors show that results? In general, these results are lost all over the figures.
Ans: The normal cell of colony formation with naringenin treatment does not include in figure 1 because naringenin does not cause normal cell mortality or colony formation as our research purpose. We only provided the cell proliferation results for indicating that naringenin does not affect the increase or decrease the number of normal cells, but affecting osteosarcoma cells as expected. Because of the effect of naringenin affecting osteosarcoma, the cancer cells were observed their colony formation during the experiment. As the purpose of colony formation, it is a standard tool to evaluate cancer cellular growth in which evaluating the stemness of particular cell population in the microenvironment as stem cells for long-living proliferation, formation of secondary tumors, and cancer recurrence with the treatment of naringenin. Therefore, colony formation only provided with osteosarcoma results with treatment that demonstrating the presence of naringenin can reduce the microenvironmental colony formation in cancer cells. With the phenotypic effects in osteosarcoma observation, naringenin demonstrated the results of cell mortality and colony formation as the experimental purpose.
Q14. Figure 2 A-B: In panel A the legend is incomplete. The meaning of AV and PI should be defined in the caption of the figure and all the plots showing the indicated % of cells in each quadrant should be included in an additional supplementary figure.
Ans: We thank the reviewer for your comments. The data have been checked and updated in the figure 2A and 2B. Additional captions of AV and PI were added in line 316 and percentage of cells in each quadrant has included in the figures.
Q15. Figure 2E: I not able to see any difference among control and treatment conditions. Could the authors explain better that differences if they really exist.
Ans: We thank the reviewer for your comments. The data have been removed and updated the labelling.
Q16. Figure 2 F-H (in fact in the caption say F, G; H is interchanged with I): could not see anything in panel F, please indicate correctly all the information.
Ans: Figure 2 and its caption have been updated to its figure labels.
Q17: Figure 6, line 385: for 8h, no “or 8h”. “The ratio of orange and green fluorescence within cells” should be graphically represented as possible, contrary that phrase (line 360-361) should be changed.
Ans: Figure 6 caption has changed for proper preposition. The ratio of orange and green fluorescence has arranged into bar figure for better understanding and corresponding to result part at line 409 to line 411.
- Galati G, Sabzevari O, Wilson JX, O'Brien PJ. Prooxidant activity and cellular effects of the phenoxyl radicals of dietary flavonoids and other polyphenolics. Toxicology. 177(1), 91-104. (2002)
- Robert SM, Ogunrinu-Babarinde T, Holt KT, Sontheimer H. Role of glutamate transporters in redox homeostasis of the brain. Neurochem Int. 73, 181-91. (2014)

Reviewer 3 Report
molecules-1415754, Naringenin induces generation of reactive oxygen species mediated endoplasmic reticulum stress, autophagy and apoptosis in human osteosarcoma
The manuscript presents a good research that fits the journal’s profile and is of interest to its readers. The methods used seem to be chosen correctly and the results interpreted accordingly.
In my opinion the title is not exact. It should correctly represent the research performed by indicating that the effects are observed on cell lines, and not in vivo. A better title would be “Naringenin induces ROS-mediated ER stress, autophagy and apoptosis in human osteosarcoma cell line”
The authors declare on row 58 that “among 121 drugs prescribed for cancer treatment, 90 are derived from herbal medicine”. The authors need to add a proper reference and to verify this statement. I don’t think this is possible when I think of all the protein kinases inhibitors, all the monoclonal antibodies. In fact, I think the number of herbal derived drugs used in clinical would be very small.
Row 61, “especially in grapefruit”. The authors should develop this idea. Mentioning just one source is not very convincing for its usefulness.
On row 63 the authors should add the doses of naringenin that can induce cytotoxicity and the cells on which is active. The same issue for the section 63-69. Add the doses and time of exposure.
Row 81, explain what ER stands for. The same for ROS
Row 107, confirm that the temperature is 34
Row 117, try to have the same style in all the paper. For example 37 °C should be 37°C. See also row 187 and check all the paper.
Section 2.2, declare the commercial source for the cell lines.
Row 145, present the concentrations of naringenin that were used. The same on row 155.
Row 246, present the exposure time
Row 258, change “Naringenin, a type of flavonoid” with “Naringenin, (2S)-4′,5,7-trihydroxyflavan-4-one”
The discussion section should be more focused on the results and on the limitations of the study. The authors should comment the high dispersion of the data and why in many cases there is no statistical difference between the control and the treated cells.
On row 467 the authors should mention the high doses used and also the pharmaceuticals problems of a drug like naringenin to reach the bone tumor tissue.
Author Response
Dear Molecules Editorial,
Thank you for your positive response regarding the publication of our manuscript in Molecules. We are glad that the reviewers find the results of this study useful to the field and greatly appreciate their valuable and helpful suggestions.
We have prepared a new revised version of the manuscript where we take into consideration the suggestions of the academic editor and reviewers. The manuscript has been carefully revised to address the suggested aspects and a point-by-point answer to the referee’s comments follow below. We used highlight (red font) to mark the changes in Word file.
We hope that these changes will be sufficient to make our manuscript suitable for publication in Molecules.
Reviewers' comments:
Comment 3:
Q1. In my opinion the title is not exact. It should correctly represent the research performed by indicating that the effects are observed on cell lines, and not in vivo. A better title would be “Naringenin induces ROS-mediated ER stress, autophagy and apoptosis in human osteosarcoma cell line”
Ans 1: We sincerely appreciate the reviewer’s comments. We have revised the title of this manuscript.
Q2. The authors declare on row 58 that “among 121 drugs prescribed for cancer treatment, 90 are derived from herbal medicine”. The authors need to add a proper reference and to verify this statement. I don’t think this is possible when I think of all the protein kinases inhibitors, all the monoclonal antibodies. In fact, I think the number of herbal derived drugs used in clinical would be very small.
Ans 2: We sincerely appreciate the reviewer’s comments. We have added a proper reference and revised the sentence.
Q3. Row 61, “especially in grapefruit”. The authors should develop this idea. Mentioning just one source is not very convincing for its usefulness.
Ans3. We sincerely appreciate the reviewer’s comments. We have provided some evidence and references in discussion.
Q4. On row 63 the authors should add the doses of naringenin that can induce cytotoxicity and the cells on which is active. The same issue for the section 63-69. Add the doses and time of exposure.
Ans 4: We sincerely appreciate the reviewer’s comments. We have provided the doses and time of exposure.
Q5. Row 81, explain what ER stands for. The same for ROS
Ans 5: We sincerely appreciate the reviewer’s comments. We have explained the ER and ROS.
Q6. Row 107, confirm that the temperature is 34
Ans 6: We sincerely appreciate the reviewer’s comments. We have confirmed that the temperature is 34oC.
Q7. Row 117, try to have the same style in all the paper. For example 37 °C should be 37°C. See also row 187 and check all the paper.
Ans 7: We sincerely appreciate the reviewer’s comments. We have revised 37°C of this manuscript.
Q8. Section 2.2, declare the commercial source for the cell lines.
Ans 8: We sincerely appreciate the reviewer’s comments. We have provided the commercial source for the cell lines.
Q9. Row 145, present the concentrations of naringenin that were used. The same on row 155.
Ans 9: We sincerely appreciate the reviewer’s comments. We have provided the doses of exposure.
Q10.Row 246, present the exposure time
Ans 10: We sincerely appreciate the reviewer’s comments. We have provided the time of exposure.
Q11. Row 258, change “Naringenin, a type of flavonoid” with “Naringenin, (2S)-4′,5,7-trihydroxyflavan-4-one”
Ans 11: We sincerely appreciate the reviewer’s comments. We have revised“Naringenin, (2S)-4′,5,7-trihydroxyflavan-4-one”
Q12. The discussion section should be more focused on the results and on the limitations of the study. The authors should comment the high dispersion of the data and why in many cases there is no statistical difference between the control and the treated cells.
Ans 12: We sincerely appreciate the reviewer’s comments. We have revised the discussion of this manuscript.
Q13. On row 467 the authors should mention the high doses used and also the pharmaceuticals problems of a drug like naringenin to reach the bone tumor tissue.
Ans 13: We sincerely appreciate the reviewer’s comments. We have mentioned the high doses used and also the pharmaceuticals problems of naringenin.

Round 2
Reviewer 1 Report
No reply
Author Response
-